# Effect of HPV Vaccination on Virus Disappearance in Cervical Samples of a Cohort of HPV-Positive Polish Patients

**DOI:** 10.3390/jcm12247592

**Published:** 2023-12-09

**Authors:** Dominik Pruski, Sonja Millert-Kalińska, Małgorzata Łagiedo, Jan Sikora, Robert Jach, Marcin Przybylski

**Affiliations:** 1Department of Obstetrics and Gynecology, District Public Hospital in Poznan, 60-479 Poznań, Poland; millertsonja@gmail.com (S.M.-K.); nicramp@poczta.onet.pl (M.P.); 2Dominik Pruski Gynecology Specialised Practise, 60-408 Poznań, Poland; 3Doctoral School, Poznan University of Medical Sciences, 61-701 Poznań, Poland; 4Department of Immunology, Pathomorphology and Clinical Immunology, Poznan University of Medical Sciences, 60-806 Poznań, Poland; 5Department of Gynecological Endocrinology, Jagiellonian University Medical College, 31-008 Cracow, Poland; jach@cm-uj.krakow.pl; 6Marcin Przybylski Gynecology Specialised Practise, 60-682 Poznań, Poland

**Keywords:** nine-valent vaccine, HPV vaccination, HPV regression, cervical intraepithelial neoplasia, HSIL

## Abstract

The introduction of human papillomavirus vaccines revolutionized cervical cancer prevention. Our research hypothesis is that HPV vaccination affects the remission of HPV in cervical swabs. We provide a prospective, ongoing, 24-month, non-randomized study in HPV-positive women. We enrolled 60 patients with positive HPV swabs from the cervix (fifty-one vaccinated with the nine-valent vaccine against HPV and nine unvaccinated). Using an enzyme-linked immunosorbent assay, we determined IgG class antibodies of HPV in the patients’ serums. Persistent HPV infection after vaccination was significantly less frequent in the nine-valent vaccinated group (23.5%) compared to the control group (88.9%; *p* < 0.001). Antibody level after vaccination was significantly higher in the vaccinated patients compared to the control group. The reactive antibody level was seen in the case of all patients in the vaccinated group and one-third of the unvaccinated group (33.3%, n = 3). The vaccination of HPV-positive patients may increase the chance of HPV remission in cervical swabs and may be a worthwhile element of secondary prevention in HPV-positive patients.

## 1. Introduction

Cervical cancer (cc) is theoretically preventable but still the fourth most common malignancy in women worldwide, with an estimated 604,000 new cases and 342,000 deaths in 2020 [1]. According to the National Cancer Registry of Poland, the standardized incidence of cervical cancer in 2019 in Poland was 12/per 100,000 women, and fortunately, the trend is downwards [2]. The latest data, covering the largest population examined in Poland so far, showed that the most common genotypes include 16, 31, 52, 66, 53 and 51 [3]. A proven risk factor for precancerous lesions and cervical cancer is persistent infection with oncogenic HPV types.

HPV is a small, non-enveloped double-strained DNA virus that belongs to the *Papovaviridae* family [4]. The HPV genome is surrounded by an icosahedral capsid that consists of two structural proteins—major protein L1 and minor protein L2 [5]. For viral replication, HPVs must deliver their genome and additional proteins into host cells [6,7]. Currently, over 200 HPV genotypes have been recognized, of which 40 are considered to be associated with genital infections. The division of HPV viruses depends on their oncogenic potential—highly oncogenic viruses, those with low oncogenic potential and those with a potential oncogenic impact. Fifteen types of HPV are categorized as high-risk (HR): 16, 18, 31, 33, 35, 39, 45, 51, 52, 56, 58, 59, 68, 73 and 82, and are associated with cervical cancer, head and neck cancer, anal and penile cancer and more. Twelve types of HPV are named low-risk (LR): 6, 11, 40, 42, 43, 44, 54, 61, 70, 72 and 81, and are responsible for genital warts and benign cervical lesions [8].

According to GLOBOCAN, nearly 70% of the population may be exposed to HPV during their lifetime [9]. The implementation of vaccination as a primary form of prevention is the most important; however, there is increasing talk about vaccination of the adult population after sexual initiation and, thus, after contact with the HPV virus. The introduction of vaccines against HPV has revolutionized cervical cancer prevention [10]. Model countries, such as Australia and Sweden, have observed a significant decrease in cervical malignancies with the population being vaccinated against HPV implemented in children and adolescents before sexual initiation, and data from others, such as New Zealand, indicate a downward trend [11,12,13,14].

Prophylactic vaccines take advantage of the fact that the HPV L1 protein is able to create virus-like particles (VLPs) when expressed in different cell types, which have a strong similarity with native virions [15,16]. They may prevent HPV infections by eliciting the production of neutralizing antibodies that bind to the viral particles and block their entrance into host cells [15,17]. Currently, there are three HPV prophylactic vaccines available commercially in Europe: Gardasil^®^4 (quadrivalent vaccine against HPV 16, 18, 6 and 11, available since 2006), Cervarix™ (bivalent vaccine against HPV 16 and 18, approved by EMA in 2007 and the FDA in 2009) and Gardasil^®^9 (non-valent vaccine against HPV 6, 11, 16, 18, 31, 33, 45, 52 and 58, available since 2014). Data from clinical trials proved their safety and good preventive effect in people infected with HPV [18]. According to data from ClinicalTrials.gov, there are studies on newly designed vaccines on different HPV genotypes—ranging from 1v to 11v.

Although therapeutic vaccines are not commercially available, more and more attention is being paid to clinical trials of them. Outcome measures concern different parameters—safety, the immune response against peptide p16_37–63, tumor response by RECIST, overall response rate (ORR), overall survival (OS), progression-free survival (PFS) and more. There are four categories of HPV treatment vaccines: live vector-based vaccines, peptide and protein-based vaccines, nucleic acid-based vaccines and whole-cell vaccines [19].

So far, the effect of vaccination on the regression of non-cancerous HPV-induced lesions, such as anogenital and laryngeal warts, has been studied. The lesions mentioned above disappeared after vaccination against HPV [20,21]. Nonetheless, no data are available describing the disappearance of HPV infection in the cervical swab after an entire course of nine-valent vaccination. HPV vaccination may also be an treatment after surgical excision of precancerous lesions [22,23,24,25]. The treatment of cervical dysplasia varies depending on the patient’s age, pregnancy status, the severity of cervical intraepithelial neoplasia and an assessment of their risk for developing CIN 3+ (CIN 3 + cervical cancer) based on their medical history. Treatment is strongly advised for all nonpregnant patients, regardless of age, with histopathologically diagnosed CIN 3. Excisional treatment is generally favored over ablative treatment [26,27]. The American Society for Colposcopy and Cervical Pathology (ASCCP) provides algorithms in which, for nonpregnant patients diagnosed with CIN 2, treatment is typically recommended. However, the decision to proceed with treatment should consider the patient’s concerns about the potential impacts on future pregnancies, balancing the patient’s maternal plans and oncological safety [28]. There are discrepancies in the literature regarding the frequency of CIN 2+ (CIN 2 + CIN 3 + early invasive cervical cancer) recurrence after the excision procedure. The recurrence rate can reach up to 20%, although it is lower in most publications [29,30,31,32,33]. Recurrence can stem from residual disease resulting from incomplete lesion removal, persistent infection, reactivation of latent HPV or acquiring a new infection post-surgery with either the same or different HPV types. The only modifiable factor we can influence after the end of surgical treatment is HPV infection.

Women after local surgical treatment for CIN are considered a target population that would benefit from vaccination to reduce the risk of CC.

Recent systematic reviews and meta-analyses concluded the reduction of recurrence of CIN after surgical treatment [23,24,25]. HPV vaccination after CIN treatment may be associated with a reduced risk of the recurrence of high-grade cervical intraepithelial neoplasia.

So far, some literature data have focused on the disappearance of early lesions [34] in the HPV-positive population; still, no PubMed publications mention the disappearance of HPV infection after the use of the nine-valent vaccine.

Despite promising data, numerous contemporary studies are either retrospective or post hoc, with study designs that did not specifically address the efficacy of vaccines. However, all existing prospective studies have consistently demonstrated a favorable impact of HPV vaccination, highlighting a reduced recurrence of CIN 1 and CIN 2 [31]. The rationale for the effectiveness of the HPV vaccine as adjuvant therapy remains unclear. Despite this, an increasing number of meta-analyses argue for its beneficial effects [35], and our work may be further evidence.

This study aims to investigate the possible implications of receiving HPV vaccination in an HPV-positive population and evaluate the clinical effectiveness of the clearance of HPV in cervical swabs.

## 2. Materials and Methods

### 2.1. Study Design

We provide a prospective, ongoing, 24-month, non-randomized study to provide the effect of nine-valent HPV vaccination in HPV-positive patients. According to data from the European Medical Agency (EMA), Gardasil^®^9 is an adjuvanted, non-infectious, recombinant, nine-valent vaccine derived from highly purified virus-like particles (VLPs) of the major capsid L1 protein of nine HPV types: 6, 11, 16, 18, 31, 33, 45, 52 and 58. The Bioethical Committee approved the study protocol (597/19). We obtained written consent for the study from all patients. All subjects reported to a private gynecological practice focused and specialized in cervical cancer diagnosis in the years 2020–2022. This is a group of patients who called for in-depth diagnostics due to a positive HPV HR test result and/or an abnormal cytological result. To verify the final diagnosis, the patients underwent colposcopy with cervical biopsy and curettage of the cervical canal. To enhance the immunological response, all subjects were offered vaccination against HPV with the nine-valent vaccine in the 0-2-6 scheme. Out of 60 patients, 51 (85%) decided to be vaccinated and 9 out of 60 (15%) did not. All patients were tested for antibody levels 6 months after the third dose of vaccination, and in non-vaccinated persons, over a similar period.

### 2.2. Inclusion Criteria for the Study

The inclusion criteria comprised:Adult females only.Non-pregnant or postpartum individuals.Subjects not undergoing immunosuppressive drug treatment.No prior vaccination with other HPV vaccines.Expressing informed written consent to participate.Consenting to proposed surgical diagnostics and potential surgical excision treatment if indicated.Having received three doses of the nine-valent HPV vaccination following the 0-2-6 month schedule.Providing blood samples at least six months after the last vaccination dose.

Exclusion criteria included:Refusal of potential treatment for squamous intraepithelial lesions.Failure to complete the full vaccination regimen.

All patients with histopathologically confirmed high-grade squamous intraepithelial lesions (CIN 2+) were treated according to the current recommendations of the Polish Colposcopic Society, with the loop electrosurgical excision procedure (LEEP), and then subjected to strict control every six months. The control group includes nine HPV-positive patients diagnosed with pre-neoplastic lesions who decided not to receive the HPV vaccine.

Figure 1 presents the scheme of the study design.

### 2.3. HPV Genotyping Test and LBC

The cervix and endocervix material were collected with a Cyto-Brush and placed in a liquid medium PreservCyt^®^ (Roche Diagnostic Systems, Meylan, France). Then, we performed PCR followed by a DNA enzyme immunoassay with a reverse hybridization line probe assay for HPV detection. To characterize HPV-positive samples, sequence analysis was performed. We used Roche Linear Array HPV Genotyping Test^®^, which identifies 32 HPV DNA of the following genotypes: 16, 18, 31, 33, 35, 39, 45, 51, 52, 56, 58, 59 and 68 (high-risk genotypes); 26, 53, 66, 70, 73 and 82 (probable high-risk genotypes); and 6, 11, 40, 42, 43, 44, 54, 61, 62, 67, 81, 83 and 89 (low-risk genotypes).

### 2.4. Colposcopy and Punch Biopsy

Colposcopy was performed for (1) abnormal cervical image under gynecological examination; (2) abnormal LBC result, i.e., ASC-US, AGC, LSIL, ASC-H, HSIL, cervical cancer; and (3) positive high-risk HPV test result. The colposcopy was performed according to the International Federation of Cervical Pathology and Colposcopy classification. In all cases, at least one cervix biopsy and cervical canal curettage was taken.

### 2.5. Specimen Collection and Handling

Six months after receiving the last vaccination, blood was drawn aseptically to the serum collection tubes (S-Monovette^®^). Afterward, the samples were centrifuged at 2000 rpm for 20 min. Supernatants (sera) were collected and frozen at −20 °C for further assays.

### 2.6. HPV Serological Measurements

We used an enzyme-linked immunosorbent assay (ELISA) kit from Creative Diagnostics (New York, NY, USA) to determine IgG class antibodies specific to human papillomavirus (HPV). The serum samples were diluted at a ratio of 1:101 in defined dilution tubes for testing. A microtiter plate coated with recombinant virus-like particles (VLP) derived from HPV types 6, 11, 16 and 18 was prepared for the assay.

Following the incubation and washing steps, we introduced the diluted samples and quality control specimens into microtiter plates, along with a peroxidase-conjugated anti-human polyclonal antibody. After another round of incubation and washing, an enzyme substrate and chromogen were added to facilitate color development. The reactions were then halted, and the optical density (OD) was measured at 450 and 620 nm, with background readings obtained at 620 nm and subtracted from the OD reading at 450 nm. The manufacturer’s specified formulation was used to calculate seropositive cut points. These cut points were established at 0.303 for HPV seronegative individuals and >0.303 for those classified as HPV seropositive.

### 2.7. Statistical Analysis

We performed an analysis in R statistical software, version R4.1.2. All tests assumed α = 0.05. Groups were characterized with median and quartile 1 and 3 or mean and standard deviation for quantitative variables, or n value and % for qualitative variables. Normality was validated with the Shapiro–Wilk test, skewness and kurtosis. Variance homogeneity was checked with Levene’s test. As appropriate, groups were compared with *t*-Student’s independent test, t-Welch’s independent test, Mann–Whitney’s U test or Fisher’s exact test. Significant differences in numerical variables between the two groups were described with a mean or median difference with a 95% confidence interval.

## 3. Results

The whole cohort included sixty HPV-positive patients: the study group included fifty-one patients vaccinated with a nine-valent vaccine against HPV, and the control group included nine non-vaccinated subjects. No difference between groups in age, obstetrics history and histopathologic outcome after LEEP-conization were found. Persistent HPV infection was significantly less frequent in the vaccinated group (23.5%) compared to the control group (88.9%, n = 8), *p* < 0.001. When only the Gardasil^®^9 type of virus (6, 11, 16, 18, 31, 33, 45, 52, 58) was considered, persistent HPV infection was significantly less frequent (11.8%, n = 6) compared to the control group (66.7%, n = 6), *p* = 0.001. Antibody level after vaccination was significantly higher in the vaccinated group compared to the control group. The reactive antibody level was seen in the case of all patients in the vaccinated group and one-third of the unvaccinated group (33.3%, n = 3). The difference was significant (*p* < 0.001). In the group of vaccinated patients, LEEP-conization of the cervix was performed in thirty-seven (72.5%) patients, and in the control group, LEEP-conization was performed in five patients (55.6%). All dependencies are shown in Table 1.

Table 2 presents subsequent clinical cases of patients, along with their level of antibodies against HPV and clinical data, such as the result of cytology or histopathological results. Table 2 also shows the HPV smear result at baseline and the vaccination course’s end. Thirty-nine out of fifty-one (76.5%) HPV-positive patients in the post-vaccination control smear and one out of nine (11.1%) subjects in the control group had no detectable HPV DNA. Twenty-four (80%) out of thirty HPV 16 (+) patients had negative cervical swabs. In six (75%) out of the eight HPV 31 (+) patients, the virus disappeared, and in four out of eight (50.0%) HPV 18 (+) patients, the virus was not detectable in the cervix.

### 3.1. Vaccinated Group—HPV Overall Positive/Negative

Patients with persistent HPV infection after vaccination did not differ from those with cleared infection in age. The level of antibodies differed depending on HPV outcome after vaccination. The proportion of histopathologically confirmed HSIL (which means an indication of LEEP-conization) and no indication for LEEP-conization was statistically different between HPV-positive and HPV-negative subjects, *p* = 0.011. The proportion of biopsy results below CIN 2 was higher in HPV-positive subjects compared to HPV-negative, as shown in Table 3. The higher level of antibodies in the HPV-positive group compared to the negative group may be due to the active replication of the HPV virus.

### 3.2. Non-Vaccinated Group—HPV Positive/Negative Considering Gardasil^®^9 HPV Types

Regarding comparing the study groups in terms of the types of viruses present in Gardasil^®^9, the relationships were as follows. Patients with persistent HPV infection after vaccination (Gardasil^®^9 HPV types) did not differ from those with cleared HPV infection in terms of age. The level of antibodies was statistically higher in patients with a positive HPV result. The proportion of no indication for LEEP-conization vs. HSIL (CIN 2 + CIN 3) was not statistically different between positive and negative outcomes, as presented in Table 3.

## 4. Discussion

The advantageous impact of HPV vaccination in safeguarding against CC by reducing the risk of HPV infection has been well-established and documented, especially when it concerns subjects before sexual initiation [36]. However, there is ongoing debate about the efficacy once HPV infection has occurred. Although no evidence-based data support HPV vaccinations or the exact timing of vaccination in the case of surgery for HPV-related lesions, many physicians recommend vaccination before conization for CIN 2+.

Our study was intended to investigate the implications of receiving HPV vaccination in the HPV-positive population and evaluating vanishing some HPV genotypes in cervical swabs. The main result indicates that after vaccination against HPV, we observed a significantly lower incidence of persistent infection compared to the unvaccinated group. This may be evidence of a more robust immune system stimulation by HPV through vaccination than spontaneous infection. We cannot relate the obtained research results to other articles because we did not find similarly designed and published works in the available databases. Greek researchers Valasoulis G. et al. presented a pilot study in which patients with low-grade cytology were vaccinated against HPV shortly after colposcopy, and they observed changes in various HPV-dependent biomarkers. According to their findings, HPV vaccination appears to significantly affect the rates of HPV 16, 18 and 31 DNA-positive infections in the population testing HPV DNA-positive for the aforementioned genotypes [37]. As we detected during our studies, in the group of vaccinated patients, the level of antibodies differed depending on HPV outcome after vaccination. In a study conducted by Mariz FC, et al., the correlation of protection with the occurrence of neutralizing or cross-neutralizing HPV IgG antibodies was not significant in recipients of the quadrivalent and bivalent vaccines. Interestingly, protective antibodies induced by the HPV vaccine may be detected up to 12 years after vaccination [38].

As per certain studies, HPV vaccination has been shown to notably decrease the recurrence of post-surgical disease in individuals with a history of prior HPV infection [39,40]. Joura et al. documented a significant reduction in the occurrence of subsequent CIN, VaIN, VIN and genital warts among women who underwent surgical treatment for HPV-related diseases following HPV vaccination [41,42]. Recently, Ghelardi et al. added insights about the mechanism of post-surgical HPV vaccination by documenting the effectiveness of the HPV vaccine in preventing recurrent disease after surgical treatment for vulvar HSIL [43]. Although our study did not concern the male population, interesting literature reports also indicate a positive impact of vaccination on the disappearance of HPV-related lesions, including external genital lesions and anogenital HPV infection in boys and men [44].

The meta-analysis from 2021 demonstrates that HPV vaccination is effective as an adjunct to surgery in preventing the risk of relapse for cervical dysplasia. The overall risk reduction of having a new or persistent CIN 2+ after surgery was 65%. The presented results might add significant clinical implications considering that the risk of relapse after surgical treatment is relatively high according to the literature, from 9% to 14% [33]. A persistently positive molecular test result may be influenced by a new infection, reinfection or autoinoculation with the HPV or immunological disorders in the course of other diseases [45]. The above statement is also supported by our results. Persistent infection or the appearance of new HPV genotypes in the smear may be associated with a new HPV infection. A systematic review and meta-analysis evaluating the efficacy of adjuvant human papillomavirus vaccination in preventing CIN 2+ after surgical excision included studies published from January 1990 to January 2019. Recurrence of CIN 2+ occurred within 6–48 months in 3.9% of women; however, recurrence was significantly lower for the vaccinated group: 1.9% vs. 5.9% (unvaccinated women). The risk of CIN 1+ was also considerably lower with HPV vaccination, occurring in 6.3% of vaccinated women compared to 9.7% of unvaccinated women. A meta-analysis conducted by Lichter K. et al. evaluated that an HPV vaccination in the setting of surgical excision for CIN 2+ is associated with a reduced risk of recurrent cervical dysplasia overall and a reduction in the risk of recurrent lesions caused by the most oncogenic strains (HPV 16, 18) [24]. Our results contradict the claims mentioned above that vaccination against HPV should be considered for additional treatment in patients undergoing surgical excision of high-grade cervical intraepithelial lesions.

In the study by Kim S. et al., in which the recurrence rate of cervical neoplasia was examined, it was concluded that after LEEP-conization in patients with CIN 2+, vaccination with a quadrivalent vaccine should be considered. In the vaccination group, 2.5% of patients developed recurrence, whereas in the non-vaccination group, 7.2%, respectively. In patients infected with HPV 16 and/or 18, five patients (2.5%) in the vaccination group (197 patients) and eighteen patients (8.5%) in the non-vaccination group (211 patients) developed a recurrent disease related to vaccine HPV types (HPV 16 or 18) after LEEP (*p* < 0.01) [46]. However, only some studies are available discussing the use of nine-valent vaccination. Our recommendations go even further; we believe that everyone, regardless of HPV infection status or history of cervical precancerous surgery, should be vaccinated. Additionally, mention should be made of patients from the STI group, who in particular suffer from HPV-related lesions not related to the cervix, such as cancer of the anal canal or nasopharynx. [47]. We believe that these patients will also benefit from vaccination.

The data from a retrospective analysis of two international, double-blind, placebo-controlled, randomized efficacy trials of quadrivalent HPV vaccine also confirm a reduced risk of any subsequent high-grade disease of the cervix. A total of 587 vaccine and 763 placebo recipients underwent cervical surgery. The incidence of any subsequent HPV-related disease revealed a 46.2% reduction in vaccination. Vaccination was also associated with a significant decrease in the risk of any subsequent high-grade disease of the cervix by 64.9%. A total of 229 vaccine recipients and 475 placebo recipients were diagnosed with genital warts or VIN, and the incidence of any subsequent HPV-related disease was 20.1 and 31.0 in vaccinated and placebo recipients, respectively (35.2% reduction) [39]. Also, post hoc analysis on the efficacy of vaccination revealed efficacy 60 days or more post-surgery for a first lesion in 88.2% against CIN 2+ and 42.6% against CIN 1+; however, the study was conducted irrespectively of prior HPV results [48].

While most studies emphasize the significant benefits of vaccinations, it is essential to acknowledge dissenting viewpoints presented in certain individual statements. Considering a degree of cross-protection against some non-vaccine genotypes genetically related to the vaccine genotypes revealed that Cervarix is more protective against HPV 31 and HPV 45 than the quadrivalent vaccine [49]. Clinical studies demonstrated comparable immunogenicity against HPV infection and cervical cancer following the administration of three doses of different prophylactic vaccines. However, the levels of anti-HPV 16 and anti-HPV 18 antibodies were notably lower after the administration of Gardasil^®^9 compared to Cervarix™ [49,50]. Notably, among various analyses, only Hildesheim et al., focusing on a population from Costa Rica, reported no significant impact of vaccination on HPV decline [51].

The obtained results are promising and give hope for the emergence of an official position of international scientific societies regarding vaccine use in the population of HPV-positive women. The high vaccination coverage of patients in the study group indicates a positive trend indicating heightened public awareness and trust in the attending physician recommending vaccination. In future studies, it would be advisable to examine HPV-infected women with the same intraepithelial lesion after HPV vaccination without conization to draw more firm conclusions. However, lesions could only concern CIN 1, because CIN 2+ lesions require the implementation of an excision procedure. Additionally, studies will be necessary to clarify whether vaccination of HPV-infected patients may increase the risk of HPV remission.

The limitation of our study is the small size of the control group. Therefore, broad generalized propositions for the population cannot be drawn. Considering this, we plan to expand the study group and further prospective studies to draw more general conclusions. The other weak point is a lack of ability to measure the antibody level at the start of the study.

The strength of the study is its statistical power according to the Open Epi tool [52]. The sample size of n = 51 in the research group and n = 9 in the control group provided 95.5% statistical power to detect significant differences between groups in the proportion of patients with positive HPV outcomes after the vaccination, assuming a two-sided 95% confidence interval and significance level of 0.05

## 5. Conclusions

In conclusion, LEEP-conization in histopathologically confirmed HSIL and a vaccination of HPV-positive patients might increase the chance of HPV remission. It is a prophylactic element worth attention for each patient and is a post-treatment effect. The observed high level of antibodies was found in a group of patients who were HPV-positive within 6–12 months after vaccination, indicating a response to active viral replication. The results of the presented analysis are preliminary, which is a significant limitation, but this also motivates us to look for additional patients to include in the next publication.

## Figures and Tables

**Figure 1 jcm-12-07592-f001:**
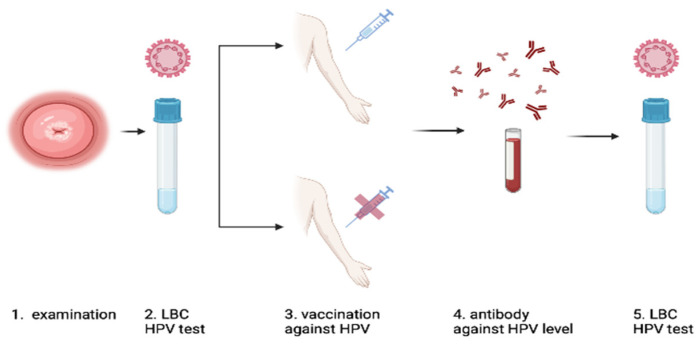
Study design; 1–2—start point; 3—vaccination in the 0-2-6 schedule; 4–5—minimum 6 months after third dose of vaccination. (1) Physical examination conducted by a gynecologist, (2) cervical swab for LBC and HPV genotyping, (3) vaccination with Gardasil^®^9 in the 0-2-6 schedule or without vaccination in the control group, (4) assessment of the level of antibodies against HPV in a study group, (5) repeated cervical smear for LBC and HPV genotyping. According to the guidelines, colposcopy and biopsy were performed if needed, and if CIN 2+ cervical pathology was found in the biopsy, LEEP-conization of the cervix was performed.

**Table 1 jcm-12-07592-t001:** Characteristics between groups.

Characteristic	Vaccinated Group	Unvaccinated Group	MD (95% CI)	*p*
N	51	9	-	-
Age, years, M ± SD	34.71 ± 6.91	33.33 ± 7.73	1.38 (−3.71; 6.46)	0.591 ^1^
Any of 32 HPV genotypes (before vac.), positive, n (%)	51 (100.0)	9 (100.0)	-	-
HPV Gardasil^®^9 type (6, 11, 16, 18, 31, 33, 45, 52, 58) before vac., n (%)				
positive, n (%)	45 (88.2)	8 (88.9)	-	
HPV Gardasil^®^9 type (6, 11, 16, 18, 31, 33, 45, 52, 58) after vac., n (%)				
Positive	6 (11.8)	6 (66.7)	-	0.001
Negative	45 (88.2)	3 (33.3)	-	
Antibody level, Me (Q1; Q3)	1.78 (1.24; 2.42)	0.17 (0.08; 0.66)	1.61 (0.67; 1.96)	<0.001 ^2^
Reactive, n (%)	51 (100.0)	3 (33.3)	-	<0.001
LEEP, n (%)				
HSIL (CIN 2 + CIN 3) + early-stage CC	37 (72.5)	5 (55.6)	-	0.431
No indication	14 (27.5)	4 (44.4)	-	

N—number; HPV—human papillomavirus; LEEP—loop electrosurgical excision procedure; HSIL—high-grade squamous intraepithelial lesion; M—mean; Me—median value; SD—standard deviation; MD—mean or median difference (positive vs. negative); CI—confidence interval; *p*—*p*-value. Comparisons were made with *t*-Student’s independent test ^1^ or Mann-Whitney’s U test ^2^ for quantitative variables and with Fisher’s exact test for qualitative variables.

**Table 2 jcm-12-07592-t002:** Distribution of HPV before and after vaccination.

Case	Group	HPV Result Before	HPV Result after Gardasil^®^9 or Observation	Antibody Level	Age	LBC Result	Biopsy Result	Leep Conization Result
1	Gardasil	6	negative	2.938	29	LSIL	HSIL	HSIL
2	Gardasil	16	negative	0.307	35	LSIL	HSIL	HSIL
3	Control	31, 53, 56, 58, 59	54, 56	0.063	21	LSIL	HSIL	HSIL
4	Gardasil	16	negative	1.681	34	LSIL	HSIL	HSIL
5	Control	16, 33, 56, 66, 68	66, 68b	1.761	26	ASC-US	HSIL	No indication
6	Gardasil	16, 18	51	2.655	36	NILM	NILM	No indication
7	Gardasil	16	negative	1.308	46	LSIL	HSIL	HSIL
8	Gardasil	18	6, 18, 56	3.348	45	ASC-US	LSIL	LSIL
9	Gardasil	52	44, 51	0.667	48	LSIL	NILM	No indication
10	Control	66	negative	0.061	39	ASC-US	NILM	No indication
11	Gardasil	16	negative	1.783	40	LSIL	HSIL	No pathology
12	Gardasil	16	negative	1.766	35	LSIL	HSIL	HSIL
13	Control	16	16	0,663	31	HSIL	HSIL	HSIL
14	Gardasil	16, 18	negative	1.654	40	ASC-US	LSIL	No pathology
15	Gardasil	31, 51	negative	0.675	30	LSIL	LSIL	No indication
16	Gardasil	16, 68, 83	negative	2.781	37	ASC-US	HSIL	HSIL
17	Gardasil	52	52	2.472	37	LSIL	LSIL	No indication
18	Gardasil	6, 16, 56, 89	16	1.454	25	ASC-US	LSIL	No pathology
19	Control	6, 39, 67	39, 51, 52	1.306	27	LSIL	NILM	No indication
20	Gardasil	53	negative	2.579	32	AGC	NILM	No indication
21	Gardasil	16	negative	2.027	32	LSIL	HSIL	HSIL
22	Gardasil	31, 56	31, 56	2.353	21	ASC-H	LSIL	No indication
23	Gardasil	16, 18, 67	56	2.952	45	NILM	NILM	No pathology
24	Gardasil	16	negative	2.221	36	LSIL	HSIL	HSIL
25	Gardasil	73	negative	0.586	29	ASC-US	NILM	No pathology
26	Gardasil	16, 53, 68, 73	56	2.888	27	ASC-H	HSIL	HSIL
27	Control	52, 62	52, 62	0.171	36	HSIL	HSIL	No pathology
28	Control	16	18	0.077	37	ASC-US	HSIL	CANCER
29	Gardasil	6, 11, 45, 51, 52, 53, 58, 61, 66, 73	45, 51, 73	2.716	24	NILM	HSIL	LSIL
30	Control	51, 52, 62, 89	51, 52, 62, 89	0.217	46	LSIL	LSIL	No indication
31	Gardasil	16, 53	negative	2.099	35	HSIL	HSIL	HSIL
32	Gardasil	16	negative	0.497	40	LSIL	LSIL	LSIL
33	Gardasil	16	negative	0.596	45	LSIL	HSIL	CANCER
34	Control	16	16	0.08	37	ASC-H	HSIL	HSIL
35	Gardasil	16, 18	16, 52	3.043	30	ASC-H	NILM	No indication
36	Gardasil	16	negative	0.972	31	HSIL	HSIL	HSIL
37	Gardasil	31, 39, 62, 70, 81, 82, CP6108	negative	0.901	45	LSIL	HSIL	No pathology
38	Gardasil	18, 31, 59	negative	1.346	24	LSIL	HSIL	HSIL
39	Gardasil	16	negative	1.176	28	HSIL	LSIL	HSIL
40	Gardasil	31	negative	1.377	41	LSIL	HSIL	HSIL
41	Gardasil	31, 53, 59	negative	2.692	29	NILM	HSIL	No pathology
42	Gardasil	16	negative	1.751	44	HSIL	HSIL	HSIL
43	Gardasil	6, 16, 62, CP6108	negative	1.081	29	ASC-US	NILM	No indication
44	Gardasil	16, 84	53	01.cze	40	AGC	CANCER	CANCER
45	Gardasil	56	negative	2.378	35	ASC-H	LSIL	HSIL
46	Gardasil	16, 31, 43, 68	negative	lut.62	35	ASC-US	NILM	No indication
47	Gardasil	16	negative	0.704	32	HSIL	HSIL	HSIL
48	Gardasil	18	negative	2.129	38	ASC-US	LSIL	HSIL
49	Gardasil	73	negative	0.733	33	NILM	NILM	No indication
50	Gardasil	73, 84	negative	lut.34	36	ASC-H	LSIL	No pathology
51	Gardasil	16	negative	1.945	34	ASC-H	HSIL	HSIL
52	Gardasil	6, 53, CP6108	negative	1.439	32	LSIL	LSIL	No pathology
53	Gardasil	52, 54, 56	negative	1.934	27	ASC-H	HSIL	HSIL
54	Gardasil	31	84	2.063	45	LSIL	LSIL	No indication
55	Gardasil	16	negative	0.626	33	ASC-H	HSIL	HSIL
56	Gardasil	16	negative	1.709	49	LSIL	LSIL	No pathology
57	Gardasil	16, 18	negative	1.802	31	ASC-H	HSIL	HSIL
58	Gardasil	51, 54, 67, 82	negative	3.132	25	HSIL	HSIL	HSIL
59	Gardasil	6, 16	negative	1.503	26	NILM	HSIL	LSIL
60	Gardasil	16	negative	2.203	35	CANCER	HSIL	HSIL

**Table 3 jcm-12-07592-t003:** Associations between HPV outcome after vaccination and selected parameters in the study group (n = 51).

Characteristic	HPV—Positive	HPV—Negative	MD (95% CI)	*p*
Any of the 32 HPV genotypes	n = 12	n = 39		
Age, years, M ± SD	35.25 ± 9.55	34.54 ± 6.02	0.71 (−5.56; 6.99)	0.811 ^2^
Antibody level, M ± SD	2.35 ± 0.78	1.64 ± 0.75	0.71 (0.21; 1.21)	0.007 ^1^
LEEP-conization result, n (%)				
HSIL (CIN 2 + CIN 3) + early-stage CC	5 (41.7)	32 (82.1)	-	0.011
No indication	7 (58.3)	7 (17.9)	-	
HPV Gardasil^®^9 type (6, 11, 16, 18, 31, 33, 45, 52, 58)	n = 6	n = 45		
Age, years, M ± SD	30.33 ± 9.11	35.29 ± 6.47	−4.96 (−10.88; 0.97)	0.099 ^1^
Antibody level, M ± SD	2.56 ± 0.66	1.71 ± 0.78	0.85 (0.19; 1.53)	0.013 ^1^
LEEP-conization result, n (%)				
HSIL (CIN 2 + CIN 3) + early-stage CC	3 (50.0)	34 (75.6)	-	0.327
No indication	3 (50.0)	11 (24.4)	-	

M—mean, SD—standard deviation, MD—mean or median difference (positive vs. negative), CI—confidence interval. Comparisons were made with *t*-Student’s independent test ^1^ or t-Welch’s independent test ^2^ for quantitative variables and Fisher’s exact test for qualitative variables.

## Data Availability

All data is available at the corresponding author.

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
