# Peer review of "Effect of HPV Vaccination on Virus Disappearance in Cervical Samples of a Cohort of HPV-Positive Polish Patients"

_jcm, 2023, doi:10.3390/jcm12247592_

Round 1

Reviewer 1 Report

Comments and Suggestions for Authors

The problem is that 72.5% of the women in the study group underwent conization. And cervical conization (and not HPV vaccination) is probably the reason why the virus cleared. HPV vaccination should prevent reinfections in these cases. You have shown that higher levels of antibodies are formed after HPV vaccination. But it cannot be concluded from these results in a heterogeneous group, where part of the women had conization and part did not, that HPV vaccination is the reason for clearance of the virus. Rather, it would be interesting to compare HPV-positive women after HPV vaccination with and without conization. The second option is to vaccinate HPV positive women without conization and monitor their PAP smears and cervical HR HPV status.

Comments:

lines 32-34

Cervical cancer (CC) is theoretically preventable but still the fourth most common malignancy in women worldwide, with an estimated 604 000 new cases and 342 000 deaths in  2020 [1]. Fortunately, the trend is downwards. According to the National Cancer Registry of Poland, the standardized incidence of cervical cancer in 2019 in Poland was 12/per 100,000 women [2].

Comment  The trend is downwards but only in developed countries and not worldwide due to the situation in developing countries. For example, according to a 2011 review (Jemal et al., CA CANCER J CLIN 2011;61:69–90), there were 529,800 new cases worldwide in 2008, which is less than in 2020. You should not mix data from the world and from a developed European country – Poland. You can leave the first sentence, but you should comment on the incidence trend in Poland.

Lines 37-40

However, he recent years that have passed in the shadow of the COVID pandemic have shown how much patients' limited access to health care also greatly impacted the diagnosis of cervical intraepithelial lesions.

Comment  Is this your opinion or is it based on statistical data? This should be stated. Or delete the sentence, it is not relevant to your study. You would do the study even if there was no COVID pandemic.

Lines 42-45

Model countries, such as New Zealand, Australia, and Sweden, have almost eliminated this malignancy by the population vaccination against HPV implemented in children and adolescents before sexual initiation [5-9].

Comment It is not true. These citations state that there are fewer HPV infections after by the population vaccination against HPV and that there is a downward trend with a good perspective for future. Wemrell writes about the degree of HPV vaccine hesitancy. These citations do not state that this malignancy has been almost eliminated in these countries. You need to rephrase the statement.

Lines 71-73

There are discrepancies in the literature regarding the frequency of CIN 2+ (CIN 2 + CIN 3 + small invasive cervical cancer which means HSIL) recurrence after the excision procedure.

Comment HSIL does not mean small invasive cervical cancer

Lines 49-91, Introduction

Commment  It is too long. There is written a lot of data irrelevant to the objectives of the study and the content of the manuscript. It needs a big reduction.

Lines 88-91

Despite encouraging data, many up-to-date studies are either retrospective or post-hoc, in  which the study design did not focus on adjuvant vaccine efficacy. All current prospective  studies have revealed a positive effect of adjuvant HPV vaccination and indicated rarer  CIN 1 and CIN 2 recurrence [23]. The rationale for the effectiveness of the HPV vaccine as adjuvant therapy remains unclear. Despite this, more and more meta-analyses argue for its beneficial effects [27], and our work is another building block.

Comment   Your vaccinations were adjuvant vaccinations only among women with conisations. Other cases of vaccinations were not adjuvant vaccinations. So it's hard to say that your work is another building block in this field.

Lines 100-101

We provide a prospective, ongoing 24-month, non - randomized study to provide the effect of 9-valent HPV vaccination in HPV – positive patients.

Comment   The effect of what?  The effect of clearance of the virus or the effect of the prevention of reinfection when the the virus was cleared due to conisation?

Lines 150-151

Colposcopy was performed for 1) abnormal cervical image, 2) abnormal LBC result, i.e., ASC-US, AGC, LSIL, ASC-H, HSIL, cervical cancer, 3) positive high – risk HPV test result.

Comment  According to the title, all patients were HPV-positive.

Lines 193-194

When only the Gardasil type  of virus (6, 11, 16, 18, 31, 33, 45, 52, 58) was considered ….

Comment  The name of the vaccine is Gardasil9.  Gardasil is 4-valent HPV vaccine.

 Lines 255-256

We cannot relate the obtained research results to other articles because we did not find similarly designed and published works in the available databases.

Comment  The problem is that 72.5% of the women in the study group underwent conization. There are quite a few articles on HPV vaccination at the time of conization. It is true that only a few of them work with Gardasil9.

Author Response

Thank You for comments and Your time. Please see the attached file. 

Reviewer 2 Report

Comments and Suggestions for Authors

In their interesting work the authors describe a possible effect of HPV-vaccination on HPV viral persistence.

The work is of great interest and actuality, and would be of interest for the readers, yet some issues need to be addressed:

1)      Please better specify in the methods section which are the inclusion criteria of patients and describe how many of them presented CIN2, CIN3, invasive cancer.

2)      The control group of patients, who did not undergo vaccination, is very small in number (9) compared to the case group, who underwent vaccination (51). The authors should enroll more controls or at least discuss the small number of controls as a weak point of the study.

3)      Antibody level of IgG class antibodies to HPV in patients’ serum and cervical swabs from cases vs. controls are compared after the vaccination of cases.

Please better specify the timelapse between the last vaccination and the analysis.

The authors should also analyze the antibody levels and the cervical swabs at the time of recruitment in the study, or at least discuss why the analyses were not performed also at time 0.

Could there be any bias? Any other factors possibly influencing viral persistence? Please consider literature reports such as:

“Gravitt PE, Winer RL. Natural History of HPV Infection across the Lifespan: Role of Viral Latency. Viruses. 2017 Sep 21;9(10):267. doi: 10.3390/v9100267.”

 4)      A sub-analysis comparing effects of vaccination in  vaccinated vs. non-vaccinated CIN 2 patients, separately from vaccinated vs non-vaccinated CIN 3 patients and separately from vaccinated vs non-vaccinated cervical cancer patients would be relevant.

Please discuss existing literature regarding the relevance of anti HPV IgG antibodies in relation to viral persistence. Please consider literature reports such as:

“Mariz FC, et al. Sustainability of neutralising antibodies induced by bivalent or quadrivalent HPV vaccines and correlation with efficacy: a combined follow-up analysis of data from two randomised, double-blind, multicentre, phase 3 trials. Lancet Infect Dis. 2021 Oct;21(10):1458-1468. doi: 10.1016/S1473-3099(20)30873-2.”

“Trevisan A, et al, The Ludwig-McGill Study Group. Correlation between cervical HPV DNA detection and HPV16 seroreactivity measured with L1-only and L1+L2 viral capsid antigens. J Med Microbiol. 2020 Jul;69(7):960-970. doi: 10.1099/jmm.0.001213. “

5)      Other reports are being published regarding the role of HPV vaccine on viral persistence also regarding LR-HPV and LR-HPV related disease: please discuss.

Please consider relevant articles such as:

“Basu P, et al. Vaccine efficacy against persistent human papillomavirus (HPV) 16/18 infection at 10 years after one, two, and three doses of quadrivalent HPV vaccine in girls in India: a multicentre, prospective, cohort study. Lancet Oncol. 2021 Nov;22(11):1518-1529. doi: 10.1016/S1470-2045(21)00453-8. “

“Giuliano AR, et al. Efficacy of quadrivalent HPV vaccine against HPV Infection and disease in males. N Engl J Med. 2011 Feb 3;364(5):401-11. doi: 10.1056/NEJMoa0909537. “

Comments on the Quality of English Language

 Extensive editing of English language required

Author Response

Thank You for Your time and comments. Please see our response in attached file.

Reviewer 3 Report

Comments and Suggestions for Authors

In the study by Pruski et al the authors describe a study of adjuvant HPV vaccination effects on subsequent cervical samples collected from previously HPV positive patients.

On the first glance it is an interesting study investigating whether giving HPV positive patients would lead to the clearance of the virus in more cases than not giving them a vaccine. However, due to somewhat chaotic design, the results cannot be readily interpreted. Critically the authors introduced confounding variables by design (treatment in some cases), or lack of sufficient initial cohort (variability of diagnosis and underlying HPV genotype at inclusion) which is further confounded by inadequate presentation of HPV genotyping data and methodological mismatches.

From 32 types assessed, only 9 are included in the vaccine. Thus only patients infected by those types at baseline could be expected to show any effect in the first place (with everything else remaining the same). Of those, only 4 types were assessed for antibody response further complicating the interpretation. The baseline antibody level status is missing, making in impossible to know whether the endpoint antibody level is due to the vaccination, underlying infection or some previous exposure.

While the study does show that most unvaccinated infections persist while in the vaccinated cohort, most infections resolved, the study design makes it completely impossible to say what was the reason since so many things ended up being different (or not clearly shown).

There are also some problems with the clarity throughout the manuscript.

More detailed comments are given in a page line format below

P2L71 apparently unfinished sentence „impacts on future pregnancies, balancing [20].“

P1-3 the introduction is possibly too long and could be streamlined. For example the first part of the  sentence “HPV vaccines are divided into prophylactic and therapeutic; however, there are few vaccination reports in the HPV-positive population” is possibly superfluous. The second part likely refers to the use of current prophylactic vaccines, not the therapeutic ones.

P3L96 suboptimal language “effectiveness of the human 95 papillomavirus 9-valent vaccine in vanishing some HPV genotypes in cervical swabs.”

P3 L105 “15th of May 2019.” Irrelevant information.

P4 L116 the inclusion criteria imply women were “(iv) not previously vaccinated with other HPV vaccines”? Likely the authors mean any previous HPV vaccine?

P4 L131 the picture could be improved by specifying the timeline. For example when were the blood samples collected and when was the second LBC sample collected. Was cytology evaluated from the same LBC sample or only HPV? It is unclear whether step 2. “LBC HPV test” refers to the initial test upon which the women were recalled (P3L107) or indeed this was another sample collected after (during?) colposcopy?  Figure 1 fails to acknowledge treatment of the lesions. Where in the timeline the treatment took place?

P4 The methods section should be streamlined. The authors describe the population as  “this is a  group of patients called for in-depth diagnostics due to a positive HPV HR test result and/or an abnormal cytological result.” and further limit the population by including only women “(vi)  agreeing to the proposed surgical diagnostics in the case of indications and possible surgical excision treatment”. The initial description is also partially misleading since apparently all women were HPV positive meaning it is likely that HPV positivity was an inclusion criteria (not “and/or” as implied earlier)

However, at this point in the manuscript it is unclear which actual diagnoses were included. For example, a woman can be HR-HPV positive while having no detectable lesions (or have low grade lesions). Inclusion criteria is written as a conditional, so such a woman only had to agree to “possible surgical treatment” while not actually requiring any treatment? Different grades of lesions are known to have different chances or persistence and regression. At this point of the manuscript it is not clear whether control and treatment groups are meaningfully comparable. Later text implies that CIN2+ was found in 72% patients given the vaccine and 55% of those not given the vaccine. It is difficult to separate the natural resolution of infections from therapeutic interventions and from vaccine effects.

P5 L141 undeclared abbreviation “Then, we performed PC followed by”

P5 L142 more detailed method description (names and or references for the assays used and possibly other technical details are required)

P5L143 if GENOTYPING was done with reverse hybridization line probe assay, what was the reason for sequence analysis (Line 143) and how was sequencing done if it was indeed done?

P5L143 “An HPV test is a quality test. It serves to identify 32 HPV….” The first sentence is unclear. The large list of types is also unclear as to which test this list is referring to.

P5 L150 what is “1) abnormal cervical image” and how was the image obtained before performing colposcopy?

P5 L153 what was biopsies in cases of ASCUS/LSIL?

P5 L165 were the VLPs provided by the manufacturer? Is there some reference or at least product name for this particular assay? Since the assay contains VLPS form 4 types, is the result available for each type separately or is the result only a combined score?

P6 L179 consider replacing “and quartile 1 and 3” with interquartile range (IQR). This would be better suited for the table 1 as well

P6 L187 there is no mention of multiple infections which could be expected in the 60 patients examined.

P6 L191 consider changing the wording to persistent and cleared infections to unambiguously differentiate cases instead of using “positive HPV test results from cervical swab after vaccination” terminology

P7 L205  Table 1 critically lacks the HPV Gardasil positivity at inclusion. Alternatively, a summary of which particular HPV types disappeared between sampling as well as whether the accompanying lesions resolved would be highly informative.

Table 1 contains both “Antibody level” and “Antibody level/cut-off” measures which is obsolete since one is directly derived from the other. It would be better to show only one. However, a note of the targeted types (6,11,16,18) is highly warranted at least as a footnote. Was the antibody level higher in patients actually positive for Gardasil types? For example were 3 seroreactive unvaccinated cases positive for those particular types?

Table 1 fails to separate individual genotype results. Ie Of the 9 un-vaccinated women 5 were actually treated by LEEP. Sill only 1 woman had HPV infection resolution after 24 months? This seems unreasonably high persistence after treatment unless de novo infections are erroneously implied as persistence of HPV.  Also is there data on HPV genotyping from the LEEP excised tissue? Which type was “removed” and is the same type found later in cytology followup specimens?

No results are shown for baseline of 24 month followup cytology results.

P8 Table 2 is not very informative and is also confusing in the middle part (Gardasil types). For example, there were 12 cases where HPV infection persisted (at least any type) and 39 cases had completely cleared all infections. However, the middle part implies that there were 6 peristent cases still positive for Gardasil types, and that there were 45 cases with negative second test.  But we are never show how many of those 45 cases were initially positive for Gardasil types in the first place.

Table 2 states ”LEEP-conization result” which implies that 5 people with persistent infection had HSIL(?) result after conization? This is wrong on several levels since biopsy/pathology results should be named using CIN nomenclature (ie CIN2+) and not cytology nomenclature (HSIL). Additionally, “no indication” should not be the result of the procedure? This nomenclature problem is found in Table 1 as well.

The supplement is mentioned  at P8L227 but not seen in the  online submission system

Comments on the Quality of English Language

minor language corrections suggested (some in the above list)

Author Response

Thank You for comments and your time. Please see the attached file.

Reviewer 4 Report

Comments and Suggestions for Authors

In the manuscript entitled “Effect of HPV vaccination on virus disappearance in a cervical swab in a cohort of 2 HPV-positive Polish patients the authors present significant data concerning the impact of Gardasil 9 (HPV nine- valent vaccine) in HPV positive patients and it is examined whether the respective vaccine contributes to the virus clearance. The manuscript is informative and outcomes are interesting, although information about the distribution of HPV genotypes in the examined patients is missing. The sample size of control group is very low in order to provide accurate results and conclusions. Nevertheless, there are indications that HPV vaccine tends to have an impact on viral clearance.  According to my opinion, some points need to be addressed.

·   The introduction is required to be improved. In particular, authors should provide more details concerning the natural history of HPV infection. Τhe description of viral DNA structure and the molecular mechanisms involved in HPV induced carcinogenesis are required to be mentioned in introduction in order to help the readers to better understand the biology of HPVs. Moreover, it is required to mention more information concerning the phylogenetic classification of HPV genome as well as it would be essential to describe in detail the members of HR and LR HPV genotypes along with the criteria that were used to classify the HPV genotypes into these two groups. (Expert Rev Mol Med. 2021;23:e19. doi:10.1017/erm.2021.18,  Virol. J. 2010;7:11.doi: 10.1186/1743-422X-7-11).

·   Authors must provide more details about the currently available vaccines, including the year of FDA approval, the name of vaccines, the  HPV genotypes that each individual vaccine protect against and the benefits of HPV vaccination considering the titers of L1 specific neutralizing antibodies compared to natural infection.  (Viruses. 2022 Dec 31;15(1):141. doi: 10.3390/v15010141, Cancer Prev Res (Phila). 2013 Nov;6(11):1242-50. doi: 10.1158/1940-6207.CAPR-13-0203)

·  Line 84: The authors mentioned that “HPV vaccines are divided into prophylactic and therapeutic; however, there are few vaccination reports in the HPV-positive population”. Currently, there are no therapeutic vaccines against HPV infection. Please revise.

·    In materials and Methods, the authors need to mention which HPV genotyping assay they used, providing the appropriate literature.

·  The authors examined 51 HPV positive vaccinated and 9 HPV positive unvaccinated patients. They report that after vaccination 39 out of 51 vaccinated patients were characterized as HPV negative, while 1 out of 9 unvaccinated patients was found HPV negative, as well.  How long after vaccination the HPV genotyping test was performed? This must be highlighted in results.

·     The authors should provide more information about the distribution of HPV genotypes among the examined samples. Moreover, they need to further analyze the characteristics of patients in both cohorts vaccinated and unvaccinated. Which HPV genotypes were they infected with? Which HPV genotypes were disappeared after vaccination? What was the intraepithelial lesion of these patients? Which HPV genotypes remain after vaccination leading to persistent infection? How many patients were diagnosed with multiple HPV infection and what authors found after follow-up HPV test. This information need to be described in detail in order to elucidate the impact of 9-valent HPV vaccine on particular HPV genotypes.

·     It is known that the majority of HPV infections are transient and cleared by the immune system, while 10–20% of infections persist latently. This must be further discussed considering the outcomes of the present analysis.

Author Response

Thank you very much for Your time and comments. Please see the attached file.

Round 2

Reviewer 1 Report

Comments and Suggestions for Authors

Dear authors,

Please read carefully your manuscript. There are some nonsenses there. I'm sorry, but 9 subjects in the control group is too few. In addition, fewer conizations were performed in the control group (55.6%) than in the HPV-vaccinated group (72.5%), which does not support your conclusions: “In conclusion, adjuvant vaccination of HPV-positive patients increases the chances of HPV remission. 

Lines 76-77

Data from clinical trails proved their safety and good preventive effect in people infected with HPV

Note: Not trails but trials. Not good preventive effect in people infected with HPV but good preventive effect against HPV in uninfected people 

Lines 620-622

Previous research suggested that the prophylactic bivalent vaccine (Cervarix™) provided greater  protection against HPV infection than the 9-valent vaccine

Note: Provide a citation.

Author Response

PART 2

Please read carefully your manuscript. There are some nonsenses there. I'm sorry, but 9 subjects in the control group is too few. In addition, fewer conizations were performed in the control group (55.6%) than in the HPV-vaccinated group (72.5%), which does not support your conclusions: “In conclusion, adjuvant vaccination of HPV-positive patients increases the chances of HPV remission.“  

Response: Thank you for pointing it out. We reworded the sentence into: „In conclusion, LEEP-conisation in histopathologically confirmed HSIL and adjuvant vaccination of HPV-positive patients might increase the chance of HPV remission”.

Lines 76-77

Data from clinical trails proved their safety and good preventive effect in people infected with HPV

Note: Not trails but trials. Not good preventive effect in people infected with HPV but good preventive effect against HPV in uninfected people 

Response: Thank you for your attention and correcting our typo. We can add a citation to support the idea that the studies were conducted on persistent HPV infection, meaning the patients were HPV (+). That is why we changed the sentence to "both infected and uninfected people" and added the citation: Crosbie, E. J., Kitchener, H. C. (2007). Cervarix–a Bivalent L1 Virus-Like Particle Vaccine for Prevention of Human Papillomavirus Type 16- and 18-Associated Cervical Cancer. Expert Opin. Biol. Ther. 7 (3), 391–396. doi: 10.1517/14712598.7.3.391.

Lines 620-622

Previous research suggested that the prophylactic bivalent vaccine (Cervarix™) provided greater  protection against HPV infection than the 9-valent vaccine

Note: Provide a citation.

Response: This is our mistake in the above sentence, we meant higher protection in the case of a bivalent vaccine compared to a quadrivalent one (Gardasil), to support which we attach a citation. Thank you for your vigilance and attention.

Reviewer 3 Report

Comments and Suggestions for Authors

The revised manuscript by Pruski et al significantly improves the manuscript. The addition of formerly missing supplementary table directly in the manuscript makes the study much easier to understand.

More detailed methodological description is also always welcome.

P6 L364 while appreciating the answers “The image of the cervix may be clinically questionable during a speculum examination by a gynaecologist.” The text of the manuscript still implies that some imaging method was applied and used when determining whether to invite the patient to colposcopy. Consider changing the text to state i.e. “abnormal cervical appearance under gynecology examination” unless actual images are routinely taken and evaluated subsequently.

P6 L367. Again appreciating the more detailed answer “Thank you for this comment. The information was included in Table, which by the incidence…” what I meant by the comment is the high likelihood that women with NILM or ASCUS or even LSIL do not necessarily always have VISIBLE lesions. The manuscript states that AT LEAST one cervix biopsy was taken in all cases even the NILM ones. So the original idea behind the question was to clarify which part of the cervix (or even parts) was sampled when no lesions were apparent (and thus easy to target). More simply what was collected when nothing abnormal was visible.

P6L380 the revised manuscript still declares no origin for the VLPs used. The authors do reply “We bought VLPs on our own but broader assay was not available; the test is a combined one.” The question remains where can those VLPs be bought? Declaring the source of VLPs makes the study replicable by other authors but more importantly allows others to verify the validity of the results. Currently it is impossible to say whether the VLPs used in the study are working or not.

Table 1 still doesn’t present the data in a meaningful way concerning the Gardasil types. The author suggests that 6 patients in the vaccinated group and 6 patients in unvaccinated group remained positive for Gardasil types. But for some cases this is not the complete picture. For example patient 19 was initially positive for 6,39,67 so would be vaccine type positive. The HPV6 was actually cleared but new infection with 52 was found afterward and this case is listed as positive for Gardasil types again. Similar is for case 28 initially positive for 16 and subsequently for 18. The presentation of the table as is might lead readers to inappropriate conclusions

Table 1 misses the footnote explaining the Me abbreviation. It is easy to mistake it for Mean, but it is apparently median value. Mean value is 0.48.

The table 1 still misses the baseline prevalence of Gardasil types.

Table 1 still misses acknowledging that unvaccinated cases with reactivity to the 6,11,16,18 VLP assay were in fact positive for those types and 2 of the 3 cleared their underlying infection with this type (cases 5 & 19).

While it is interesting to dig into the table 2, it might be more appropriate to put such summaries in table 1 where they would be expected as result.

The added limitation of the study as well as  the acknowledgment that persistent infections may In fact be newly acquired infections (p13 L575) are welcome additions to the manuscript.

Author Response

Thank you very much for your attention and valuable comments. 

The revised manuscript by Pruski et al significantly improves the manuscript. The addition of formerly missing supplementary table directly in the manuscript makes the study much easier to understand.

More detailed methodological description is also always welcome. 

P6 L364 while appreciating the answers “The image of the cervix may be clinically questionable during a speculum examination by a gynaecologist.” The text of the manuscript still implies that some imaging method was applied and used when determining whether to invite the patient to colposcopy. Consider changing the text to state i.e. “abnormal cervical appearance under gynecology examination” unless actual images are routinely taken and evaluated subsequently.

Response: Thank you very much for appreciating the efforts to improve the text. We really want it to be readable. We agree that the quoted sentence will sound more favorable, that is why we reworded the text into: “Colposcopy was performed for 1) abnormal cervical image under gyneacological examination, 2) abnormal LBC result, i.e., ASC-US, AGC, LSIL, ASC-H, HSIL, cervical cancer, 3) positive high – risk HPV test result.”.

P6 L367. Again appreciating the more detailed answer “Thank you for this comment. The information was included in Table, which by the incidence…” what I meant by the comment is the high likelihood that women with NILM or ASCUS or even LSIL do not necessarily always have VISIBLE lesions. The manuscript states that AT LEAST one cervix biopsy was taken in all cases even the NILM ones. So the original idea behind the question was to clarify which part of the cervix (or even parts) was sampled when no lesions were apparent (and thus easy to target). More simply what was collected when nothing abnormal was visible. 

Response: Yes, we agree that the image of the cervix, even after the acetic acid test and the Schiller test, does not always indicate a suspicious place. In the above situation, the cervical biopsy was performed from the transformation zone (TZ).

P6L380 the revised manuscript still declares no origin for the VLPs used. The authors do reply “We bought VLPs on our own but broader assay was not available; the test is a combined one.” The question remains where can those VLPs be bought? Declaring the source of VLPs makes the study replicable by other authors but more importantly allows others to verify the validity of the results. Currently it is impossible to say whether the VLPs used in the study are working or not.

Response: VLPs were purchased in the entire kit. In the Methodology section, we have provided the manufacturer: Creative Diagnostics (New York, USA), so that other researchers will also be able to perform similar tests.

Table 1 still doesn’t present the data in a meaningful way concerning the Gardasil types. The author suggests that 6 patients in the vaccinated group and 6 patients in unvaccinated group remained positive for Gardasil types. But for some cases this is not the complete picture. For example patient 19 was initially positive for 6,39,67 so would be vaccine type positive. The HPV6 was actually cleared but new infection with 52 was found afterward and this case is listed as positive for Gardasil types again. Similar is for case 28 initially positive for 16 and subsequently for 18. The presentation of the table as is might lead readers to inappropriate conclusions 

Response: To avoid any understatements, we have added a line indicating the number and percentage of patients with HPV present in Gardasil9. We hope that Table 1 is more readable now.

Table 1 misses the footnote explaining the Me abbreviation. It is easy to mistake it for Mean, but it is apparently median value. Mean value is 0.48. 

Response: We added the explanation.

The table 1 still misses the baseline prevalence of Gardasil types. 

Response: We have added a line indicating the number and percentage of patients with HPV present in Gardasil9. In vaccinated and unvaccinated group is similar percentage of Gardasil9 HPV type – 88.2% and 88.9%, respectively.

Table 1 still misses acknowledging that unvaccinated cases with reactivity to the 6,11,16,18 VLP assay were in fact positive for those types and 2 of the 3 cleared their underlying infection with this type (cases 5 & 19). While it is interesting to dig into the table 2, it might be more appropriate to put such summaries in table 1 where they would be expected as result.

Response: If necessary, we will also add a line with positive test results for HPV 6,11,16,18, but we think this may cause chaos. Other reviewers have not indicated this, so we will wait for the opinion of the Academic Editor.

The added limitation of the study as well as the acknowledgment that persistent infections may In fact be newly acquired infections (p13 L575) are welcome additions to the manuscript.

Response: Thank you, we underlined it in the discussion. “A persistently positive molecular test result may be influenced by a new infection, reinfection or autoinoculation with the HPV or immunological disorders in the course of other diseases [45]. The above statement is also supported by the our results. Persistent infection or the appearance of new HPV genotypes in the smear may be associated with a new HPV infection.”

Reviewer 4 Report

Comments and Suggestions for Authors

The authors reply to my comments sufficiently. However, the study group remains heterogeneous and the sample size of control cohort is still low. According to my opinion cervical conization might be the reason of virus clearance and not HPV vaccination. It would be essential to examine HPV positive women with the same intraepithelial lesion after HPV vaccination without conization in order to lead in more definite conclusions. Considering the results derived from the present analysis it is not clear whether adjuvant vaccination of HPV-positive patients may increase the chances of HPV remission. 

Author Response

Thank you for appreciating our efforts. We are aware of the limitation of a small control group. In the current publication, we are not able to change its number, but we emphasized it in the Limitation of the study. We are currently working on a publication with a group of 300 patients in the control group. However, the planned publication will not include information on HPV antibody levels.

We agree that the impact of excision surgery, among others, LEEP-conization of the cervix has an impact on the final diagnosis in patients and the disappearance of the disease. Therefore, we emphasize in the Conclusions section: "In conclusion, LEEP-conclusion in histopathologically confirmed HSIL and adjuvant vaccination of HPV-positive patients might increase the chance of HPV remission."